# Free-Space Optical Data Receivers with Avalanche Detectors for Satellite Downlinks Regarding Background Light

**DOI:** 10.3390/s22186773

**Published:** 2022-09-07

**Authors:** Dirk Giggenbach

**Affiliations:** German Aerospace Center (DLR), D-82234 Wessling, Germany; dirk.giggenbach@dlr.de

**Keywords:** optical avalanche photodiode receiver frontend, RFE, InGaAs-APD, bias voltage control, temperature control, Q-factor, background light, free-space optical communication, FSO

## Abstract

Data receiving frontends using avalanche photodiodes are used in optical free-space communications for their effective sensitivity, large detection area, and uncomplex operation. Precise control of the high voltage necessary to trigger the avalanche effect inside the photodiode depends on the semiconductor’s excess noise factor, temperature, received signal power, background light, and also the subsequent thermal noise behavior of the transimpedance amplifier. Several prerequisites must be regarded and are explained in this document. We focus on the application of using avalanche photodiodes as data receivers for the on/off-keying of modulated bit streams with a 50% duty cycle. Also, experimental verification of the performance of the receiver with background light is demonstrated.

## 1. Introduction

Avalanche photodiode (APD) detectors, with indium-gallium-arsenide semiconductor technology (InGaAs), together with low-noise transimpedance amplifiers (TIA) operating at 15xx nm wavelength, are becoming the standard receiver frontend (RFE) technology for optical inter-satellite links (as with mega-constellation satellite networks), as well as in space-to-ground communication [1,2,3,4,5,6,7,8]. APDs have been in use in low-cost fiber communication links (for up to 10 Gbps data rates) since their internal photo–electron gain allows for extended distance reach when compared to less sensitive ~10 dB (and more conventional) positive intrinsic negative photodiode (PIN) receiver technology [9]. While a constraint-linked budget in fiber links can be compensated for through intermediate optical amplifiers, this is not possible in long-range free-space optical communications (FSO). Therefore, APD-RFEs are the preferred opto-electric frontend for optical space links with intensity modulation and direct detection (IM/DD). Most relevant is the simple On/Off-Keying (OOK) modulation, i.e., a digital ‘one’ is coded as a signal pulse. The foundations for such downlink technology were laid by the GEO-downlinks from OPALE, onboard ESAs Artemis satellite in the SILEX demonstration [10] and the OICETS experiments from 2005–2009, with silicon-based APDs operating at 8xx nm wavelength at that time [10,11,12,13]. In current implementations, the wavelength range of around 1550 nm has evolved to become the most useful due to the re-use of terrestrial fiber communication components (foremost, the efficient erbium-doped fiber amplifier, but also the range of wavelength channels, data-modulators, and detectors). Also, the high achievable data rates and other methods for increasing the sensitivity (like optical pre-amplification or coherent modulation) are desirable. Here, optical space communications follow terrestrial systems [14], but these often require a higher implementation effort, such as with adaptive optics for the single mode fiber coupling of an atmospherically distorted optical field [15]. Bulk AP-detectors do not require such sophisticated techniques and allow focal illumination with an atmospherically distorted light field. This typically works for up to 10 Gbps and slightly more, whereas, for higher data rates, only the coupling of this into a single-mode fiber is required. Besides pure communication, optical data transmission allows for the application of other photon-based techniques, like the exploitation of quantum effects for key-distribution transmission [16]. Current implementations of space–ground links employ component technology from dense wavelength division multiplexing (DWDM) with bulk optical detectors at optical ground stations (OGS) [17,18,19].

A variety of environmental and orbital parameters influence the short- and long-term received optical power in long-range mobile FSO systems, such as satellite up- and down-links [20]. The reasons for changes in the mean received power–besides being blocked by cloud [21]–are the intermittent increase in atmospheric attenuation (haze, precipitation) [22], varying free-space losses (caused by changes in link-range), beam pointing errors [23], and scintillations caused by the atmosphere’s index of refraction turbulence (IRT) [24,25,26]. Altogether, the mean received optical power can change by orders of magnitude over very short time frames. Sophisticated data-based electronic techniques, like adaptive coding and rate adaptation, might be able to cover these steep and fast variations. However, physical APD-RFE parameters, like diode bias voltage, limiter decider threshold, and receiver bandwidth, must also be constantly adopted to ensure optimum receiver performance. Specifically, the bias voltage is not only a function of temperature but also of instantaneous received optical power, dark current, and foremost, the background light [27,28,29,30,31,32]. The strength and velocity of such parameter variations in a typical low-Earth orbit (LEO) to ground FSO scenario (~500 km circular orbit) can be estimated as follows [25,33]:Received signal power changes due to distance variations via orbital movement from a 5° elevation to zenith of ~12 dB. This change happens in minutes;Additional signal power variation is introduced by an increased atmospheric attenuation at low elevations, adding up to ~6 dB of loss when close to the horizon, depending on OGS location;Background light, in daytime, near to the horizon is stronger than at zenith by up to ~10 dB, with a variation speed similar to those in the atmospheric attenuation changes;Received power scintillations (due to atmospheric IRT) and pointing fading (caused by finite beam direction control) adds another ~6 dB or more, on the timescale of milliseconds;Furthermore, the fast angular movement of a LEO satellite across the sky (up to more than 1°/s) will lead to additional fading from unwanted miss-pointing.

Data RFE performance can be determined by its required average energy-per-bit to achieve a given electrical signal quality, stated as, e.g., in SNR, Q-factor, or bit error ratio (BER). Here, a typical value for InGaAs-APD-RFEs operating within a near 1550 nm wavelength is around 500 photons per bit (Ppb) for BER = 10^−9^ and better. Overviews of practically achieved RFE sensitivities can be found in [34] on p. 330 and in [35] (Table 3). An even higher sensitivity has been reported by an alternative APD semiconductor material, HgCdTe, and commercial devices for FSO communication applications might soon be demonstrated [36,37].

In a typical high-speed optical receiver setup, the light-sensitive photodiode generates primary photoelectrons when illuminated by background or signal light power. The reverse biasing of the diode with *U_R_* causes an E-field that accelerates the primary photoelectrons, which, when colliding with other electrons, excite secondary and possibly ternary signal electrons. This electron avalanche acts as an internal diode amplification of the photocurrent, which is then converted into a voltage signal via transimpedance, *R_TI_*, filtered according to its required signal bandwidth, and its binary is decided by a limiting circuit when data reception takes place (see Figure 1).

The RFE features a variable *U_R_* that is adjusted according to APD temperature (monitored by a temperature sensor, *S_Temp_*), total APD current, as monitored by *I_APD_*, optical received power, *P_Rx_*, (cannot be monitored directly without an extra power sensor), and the electrical signal amplitude, as monitored via the received signal strength indicator, RSSI. The Rx filter bandwidth, *B,* is adjusted to a 3 dB cutoff at roughly half the data rate, *r*. To eliminate the influence of electronic offsets, AC coupling is introduced before the limiter. The latter’s threshold voltage, *U_Thr_*, is also optimized via RSSI according to the asymmetric noise distributions of the APD-RFE. The system performance can finally be evaluated via a bit error ratio tester.

For an RFE with a variable bandwidth/data rate, it would be most beneficial in terms of sensitivity to adapt the feedback resistance, *R_TI_*, to the required bandwidth. However, this is not feasible with practical high-speed (>1 Gbps) systems employing integrated circuits with a fixed feedback structure. Thus, the reduction of noise through the spectral filtering by the Rx filter offers only a part of the possible SNR optimization in a variable data rate RFE.

The remainder of this document is organized as follows. Section 2 summarizes the basic relationships between the photodetectors and APD performance modelling. Section 3 shows the dependency and optimization of the avalanche multiplication factor, *M,* for CW illumination, and Section 4 describes the theoretical RFE performance for the On/Off-keying of NRZ data modulation, with general sensitivity modelling between the two boundary cases: thermal and shot–noise limit. Section 5 introduces the influence of background light on the receiver’s performance, where an analytical optimization for the multiplication factor is presented. A comparison with measurements from an exemplary APD-RFE are given. Section 6 presents the summary and conclusions. An Appendix A provides several helpful relations for RFE assessment.

## 2. Gain and Noise in Avalanche Photodiodes

In this section we consider the APDs sensitivity for an unmodulated or continuous wave (CW) optical signal illuminating the photosensitive detector area.

The received optical power, *P_Rx_*, in the form of photons impinging onto the detection area of a signal receiver, is the exciting of photoelectrons from the semiconductor’s valence-band, via the detector material’s responsivity, *R*. This responsivity depends on signal wavelength, *λ*, and the semiconductor’s conversion efficiency, *η*. An additional multiplication gain, *M*, is achieved in the APDs by an electric field inside the detection volume, which accelerates the primary photoelectrons. These again strike out further (secondary) electrons in the multiplication region. This way, the signal current is increased by an internal multiplication factor of *M*:(1)$$I_{Sig}=M\cdot P_{Rx}\cdot R=M\cdot P_{Rx}\cdot \left(\eta \cdot \frac{q\lambda }{hc}\right)$$

*c* = 2.998·10^8^ m/s (speed of light in vacuum)

*h* = 6.626·10^−34^ Ws^2^

*q* = 1.602·10^−19^ As

*R*, Responsivity [A/W], for λ = 1550 nm is *R* = *η*·1.250 A/W

*η*, quantum efficiency of the diode’s detection region (typ. ~0.75 to 0.9)

The reception of a static power level is subject to variations in the photon-excited photoelectrons due to stochastic conversion variations and the variations of the photon-arrival probability itself. In APDs (as well as in most electron multiplication devices), the multiplication effect itself is subject to variations, resulting in additional noise effects. Expressions for the accordant distributions have been derived by McIntyre [38,39], confirmed experimentally by Conradi [40], and reviewed and simplified by Webb [41]. The signal-to-noise power ratio is derived for CW illumination as the ratio of the square of the signal current (from constant illumination, where we neglect the influence from level-offsets by background light and dark currents) to the sum of shot-noise and thermal-noise variances in the current (*σ_s_*^2^ and *σ_t_*^2^), where shot-noise variance *σ_s_^2^* in [*A^2^*] is calculated from any flowing current *I* $\sigma_{S}{}^{2}=2q\cdot I\cdot B$, and thermal noise density *i_t_* [$A/\sqrt{Hz}$] defines the thermal noise variance $\sigma_{t}{}^{2}=i_{t}{}^{2}\cdot B$.
(2)$$\begin{array}{c} \frac{S}{N}={\left(\frac{I_{sig}}{I_{n}}\right)}^{2}=\frac{I_{sig}{}^{2}}{\sigma_{s}{}^{2}+\sigma_{t}{}^{2}}=\\ =\frac{{\left(P_{Rx}RM\right)}^{2}}{B\cdot \left\{2q\left[I_{du}+\left(P_{Rx}RF_{sm}+\left(I_{dm}+R\cdot P_{bgl}\right)F_{dm}\right)\cdot M^{2}\right]+i_{t}{}^{2}\right\}} \end{array}$$

With the APDs multiplication factor *M*, the unmultiplied dark current component is *I_du_*, multiplied dark current component is *I_dm_*, responsivity of the detector material is *R*, excess noise factor for signal current is *F_sm_*, excess noise factor for the multiplied dark current is *F_dm_*, and observation bandwidth is *B*. *I_dm_* is given through the biasing of the detector material itself, but is constant around the operating bias point, *U_APD_*; it will however generate a dark-current component, which is variable by its multiplication with *M*.

The thermal noise current density of the succeeding amplifier stage *i_t_* can practically be found in data sheets, or is stated with the load resistor, *R_L_*, noise figure, and other constants (compare Section A.2). False background light power, *P_BGL_*, can illuminate the detector and is added to the multiplied dark current. By making *M* large, the *M^2^* term in the denominator of (2) becomes larger than the thermal noise of the amplifiers, improving the SNR towards shot noise limitation. However, the APDs excess noise factor, *F_A_*, will cause a local maximum here. We can see that the performance of an APD is defined by its dark current, the excess noise factors, and an optimum *M*.

The excess noise factors are often simplified by equaling *F_A_ = F_dm_ = F_sm_*, and it quantifies the additional noise caused by the fluctuation of the avalanche multiplication process [41]:(3)$$F_{A}=\frac{\langle M^{2}\rangle }{{\langle M\rangle }^{2}}$$

*F_A_* depends on the ratio of electron-hole generation *k_A_ = α_h_/α_e_* in the specific APD design and material (0 *< k_A_ <* 1), by defining *k_A_* as the smaller of the ratios between holes vs. electrons. *F_A_* has been derived as
(4)$$F_{A}\left(M\right)=k_{A}\cdot M+2\left(1-k_{A}\right)+\left(k_{A}-1\right)/M$$

For current InGaAs-APDs, we find *F_A_* typically ranging from 3 to 5 for *M* = 10, and from 4 to 8 for *M* = 20.

Different approximations have been suggested for (4) to allow for the further analytical evaluation of APD performance. The exponential approximation
(5)$$F_{A}\approx M^{k_{A}{}^{0.355}}$$
models well for a smaller *M*, and the exponent can be adopted to other ranges of *M*. Furthermore, a linear approximation is in use [42]:(6)$$F_{A}\approx 2-2k_{A}+k_{A}M$$

From the comparison in Figure 2, we find that both approximations provide a good fit for 10 *< M <* 20.

## 3. Optimum Multiplication for CW Illumination

Depending on *P*_0_, the optimum multiplication factor *M_opt,CW_* can be found for the CW case by solving *d(SNR)/dM =* 0.

With the exact relation or the linear approximation for *F_A_*, further simplifications would be necessary to solve for *M_opt,CW_*. However, with the exponential approximation (5), we can derive a closed-form solution for the optimum multiplication factor with CW-illumination:(7)$$M_{opt,\;CW}={\left(\frac{2q\;I_{dm}+i_{t}{}^{2}}{x\cdot q\;\left(RP_{Rx}+RP_{BGL}+I_{dm}\right)}\right)}^{\frac{1}{2+x}}$$
with $x=k_{A}{}^{0.355}$.

*M_opt,CW_* is a function of the received power and further RFE parameters, but is independent of the observation bandwidth *B*.

For typical InGaAs-APDs and TIAs (using for *i_t_* a typical 2.1 pA Hz^−0.5^), we find 10 *< M_Opt_ <* 30 (see Figure 3).

We assume here that the excess noise factors for signal electrons and multiplied dark current electrons are equal; this assumption causes negligible errors when the signal current is dominating the total avalanche current.

### Dependency of M from Bias Voltage and Temperature

The electric field induced through the reverse voltage, *U_R_*, causes the electron-hole avalanche, starting with M = 1 at a minimum operating voltage (typically one-third of the breakdown voltage, *U_BD_*). Beyond ~0.5·*U_BD_*, the multiplication increases according to (8) up to a pole at *U_BD_* (see Figure 4). When no optical data power is applied, only the APDs dark current is multiplied, and the *U_BD_* is then typically defined as the voltage, in which the reverse current exceeds 100 µA.

The relationship between bias voltage, *U_R_*, and gain, *M,* is related by a simple rational term [42,43]:(8)$$M\left(U_{R}\right)=\frac{1}{1-{\left(\frac{U_{R}-I_{APD}\cdot R_{S}}{U_{BD}}\right)}^{n}}$$

The term $I_{APD}\cdot R_{S}$ in (8) indicates the voltage lost across the additional series resistance, *R_S_*, of the photodiode (resistance of contacts and nondepleted semiconductor region). This resistance is typically around 1 kΩ, and the voltage-term is thus negligible for data communication. But it needs to be regarded in a situation with high optical power, especially since *I_APD_*, again, is a function of the multiplication factor. Exponent *n* can be adopted to an improved fitting to measured behavior. Typically, its value is slightly above 1.

Figure 4 compares the measured *M* of an InGaAs-APD-RFE (derived from RFE #7 of [35]) with the fit, according to (8). We understand that for this type of APD, the *U_R_* is typically 4% to 8% below the breakthrough.

*U_BD_*, as well as *U_R_*, reduces along with the temperature, and can be approximated by a linear fit using the temperature coefficient *ρ_T_ =* Δ*U_R_/*Δ*T* [44]
(9)$$U_{R}=U_{R,ref}+\rho_{T}\cdot \left(T-T_{ref}\right)$$

We find the required operating voltage, *U_R_*, for a certain *M* by solving (8) with (9) (neglecting the series resistance), as illustrated in Figure 5:(10)$$U_{R}=\left[U_{BD,ref}+\rho_{T}\cdot \left(T-T_{ref}\right)\right]\cdot {\left(1-\frac{1}{M}\right)}^{\frac{1}{n}}$$

We now can use the optimum *M* from (7) to describe the complete APD voltage control loop for CW illumination, including the temperature compensation.

## 4. Uncoded BER of APD OOK Receivers

We consider the bit decision of an OOK data stream with an APD-RFE, including its noise processes. Figure 6 shows a typical pseudo-random bit sequence (PRBS) with 100 Mbps detected with an APD-RFE. Regard the stronger shot-noise caused by the OOK signal during the reception of the binary on bits. Further information on direct detection receivers can be found in [45,46,47,48,49,50,51,52].

Noise is assumed to be Gaussian-distributed with a different noise sigma for 1 s and 0 s (Figure 7). A detection threshold current level, *I_Thr_*, between the curves around the 0 s and 1 s, decides for a binary 0 and 1. Decisions to the wrong side from *I_Thr_*, will lead to bit errors. Their probability is linear to the ratio of the stippled areas *A + B* related to the total curve-area.

The areas on the wrong side of *I_Thr_* are minimized with an optimum decision threshold level. For an analytically applicable approximation, *I_Thr_* is defined by the intersection of both curves, which leads to
(11)$$I_{Thr}=\frac{\sigma_{0}\cdot \langle s_{1}\rangle +\sigma_{1}\cdot \langle s_{0}\rangle }{\sigma_{0}+\sigma_{1}}$$

The quality factor *Q* can then be derived as
(12)$$Q=\frac{\langle s_{1}\rangle -\langle s_{0}\rangle }{\sigma_{1}+\sigma_{0}}$$

Assuming *<s*_0_*> =* 0 (no offset), by integrating over the tails of the two Gaussian distributions, we get the bit error probability, *p_BE_.*
(13)$$p_{BE}=\frac{1}{2}\cdot erfc\left(\frac{1}{\sqrt{2}}\cdot \frac{\langle s_{1}\rangle }{\sigma_{0}+\sigma_{1}}\right)=\frac{1}{2}\cdot erfc\left(\frac{1}{\sqrt{2}}\cdot Q\right)$$
where $\langle s_{1}\rangle =M\;R\;{\hat{P}}_{Rx}$ is given by the optical power during an on-symbol, which is, again, twice the average power $\langle P_{Rx}\rangle$. The noise current during “0” and “1” consists of the same thermal noise current:(14)$$\sigma_{t}=i_{t}\cdot \sqrt{B}$$
and the shot noise from background light and dark currents is
(15)$$\sigma_{s,o}=\sqrt{B\cdot 2q\left[M^{2}F_{A}\left(I_{dm}+R\cdot P_{bgl}\right)+I_{du}\right]}$$
where during a binary “1”, we see additional signal shot noise:(16)$$\sigma_{s,1}=\sqrt{B\cdot 2q\left\{M^{2}F_{A}\left[R\;\cdot \left(2\langle P_{Rx}\rangle \right)+I_{dm}+R\cdot P_{bgl}\right]+I_{du}\right\}}$$

*Q* then becomes
(17)$$Q_{st,APD}=\frac{M\;R\;\cdot \left(2\langle P_{Rx}\rangle \right)}{\sqrt{\sigma_{s,0}{}^{2}+\sigma_{t}{}^{2}}+\sqrt{\sigma_{s,1}{}^{2}+\sigma_{t}{}^{2}}}$$

When neglecting the small unmultiplied dark current *I_du_*, we find
(18)$$\begin{array}{ll} Q_{st,APD} & =\frac{M\;R\;\cdot \left(2\langle P_{Rx}\rangle \right)}{\sqrt{B}\cdot \sqrt{2qM^{2}F_{A}\left(RP_{bgl}+I_{dm}\right)+i_{t}{}^{2}}+\ldots }\\ & \frac{}{\ldots +\sqrt{B}\cdot \sqrt{2qM^{2}F_{A}\left(R\left(2\langle P_{Rx}\rangle +P_{bgl}\right)+I_{dm}\right)+i_{t}{}^{2}}} \end{array}$$

### Sensitivity Estimation without Background Light

We find a general simplified (no *P_BGL_*, no *I_d_*) receiver sensitivity formula in terms of the Q-factor, with signal power during the reception of a binary 1 being $P_{1}=2\langle P_{Rx}\rangle$, and the signal level during a binary 0 being is $P_{0}=0$. We regard the noise during the binary 0 and 1 as ($\sigma_{0}=\sigma_{thermal}$ and $\sigma_{1}=\sqrt{\sigma_{shot-1}{}^{2}+\sigma_{thermal}{}^{2}}$) and photon energy $E_{Ph}=hc/\lambda$, and assume equal distribution of 1s and 0s in the bit-stream:(19)$$\begin{array}{ll} Q_{APD} & =\frac{M\;R\;\cdot \left(2\langle P_{Rx}\rangle \right)}{\sigma_{t}+\sqrt{\sigma_{shot-1}{}^{2}+\sigma_{t}{}^{2}}}=\\ & =\frac{\sqrt{8r}\cdot M\;R\cdot \langle N\rangle \cdot E_{Ph}}{i_{n}+\sqrt{4qM^{2}F_{A}R\cdot \langle N\rangle \cdot E_{Ph}\cdot r+i_{n}{}^{2}}} \end{array}$$

From *Q,* we can derive *p_BE_* via (13).

In one extreme, shot noise is negligible vs. thermal noise, and the RFE operates purely in the thermal-noise limit (*TNL)*:(20)$$Q_{TNL}=\frac{R\;\cdot \langle P_{Rx}\rangle }{\sigma_{t}}=\frac{R\;\cdot \langle P_{Rx}\rangle }{i_{t}\cdot \sqrt{B}}=\frac{\sqrt{2r}\cdot R\;\cdot E_{Ph}}{i_{t}}\cdot \langle N\rangle$$

Thus, *Q* is linear to *<N>*, and to achieve a constant *Q,* the bitwise sensitivity will decrease by $\sqrt{r}$, i.e., larger bandwidth shall be beneficial in terms of sensitivity. This relation however holds only as long as the receiver’s amplifier is not adopted by *B*. But an adoption might be required to achieve a certain data rate, which at the same time means choosing the highest possible *R_TI_*. Then again, the data rate advantage of *TNL* practically will be compensated, compare Figure A2.

In the other ideal extreme of shot-noise limit (*SNL)*, multiplication *M* cancels, excess noise *F_A_* becomes one, and thermal noise *i_t_* is negligible vs the shot-noise:(21)$$Q_{SNL}=\frac{M\;R\;\cdot 2\langle P_{Rx}\rangle }{\sigma_{s,1}}=\sqrt{\frac{2R\;\cdot \langle N\rangle \cdot E_{Ph}}{q}}=\sqrt{2\eta \langle N\rangle }$$

So in *SNL,* the sensitivity in *Ppb* is independent from the data rate, and the run of *Q* now corresponds linear to $\sqrt{\langle N\rangle }$ ([45] ch. 4.5.2).

The results here always assume Gaussian noise distributions and the presence of thermal noise during a zero bit. This theoretically is not true for an ideal photon-counting receiver without BGL or dark current, and becomes obvious when the received number of photons is low. There will be no noise during zero, thus, the decision threshold will be at nearly zero, and furthermore, the photon arrival probability will be Poisson-distributed, not Gaussian anymore [53]. The sensitivity numbers calculated with (20) and (21), therefore, deviate from ideal statistics when the numbers of photon per bit becomes very low. In our regime of dozens of photons per one bit, however, the preceding formulas provide a sufficiently accurate approximation.

The above findings can be compared on logarithmic scales for the received power, as shown in Figure 8, Figure 9 and Figure 10, employing real-world APD parameters from Table A1. In Figure 8 and Figure 9, the data rate is varied while the TIAs noise density stays constant. Thus, only the reception filter is adopted by the data rate. We find that when *M* = 25, this is an advantageous fixed value for 1 Gbps, when no *I_d_* or *P_BGL_* are present (compare Figure 11), which is, however, only an idealized case.

The sensitivity runs of the different RFE types become steeper, regarding SNL via APD to TNL, and typically a 10 dB sensitivity step is seen for the same BER in between these technologies (Figure 8, comparing the plots of the same line style). However, going from PIN to APD is only a moderate technology step, while building and operating a SNL (photon counting) receiver at 15xx nm wavelength requires much higher complexity and expenditure [54]. Figure 9 plots the BER sensitivity in photons per bit for a fixed M = 25 and a TIA covering rates up to 10 Gbps. Since the TI amplifier’s noise density is assumed constant here (2.1 pA/sqrt(Hz) for all data rates, and only the bandwidth of the separate reception filter is adopted by the data rate, then the sensitivity is rate dependent.

This assumption, however, is impractical for many systems since the selection of the TIA component would then depend on the highest employed bandwidth (here 10 Gbps), which would mean unnecessary high noise density values for the lower rates (1 Gbps or 100 Mbps). In other words, the TIAs noise density in a data rate-optimized RFE setup must follow its bandwidth. As a simple explanation for this behavior, the amplification of the TIA can be chosen as being higher to offer a better SNR, but at the same time, the bandwidth is reduced through *1/R_TI_*.

So, in another setup, one adopts the bandwidth of the TIA (and thus its *R_TI_*) with the sqrt(B) characteristic of the current noise density, *i_n_*, as shown in Figure A2 and in Table A1 for a 500 MHz bandwidth. For data rates of 100 Mbps/1 Gbps/10 Gbps, the noise densities are 0.66 pA Hz^−0.5^, 2.1 pA·Hz^−0.5^, and 6.6 pA Hz^−0.5^, respectively. This results in TIA-adopted sensitivity for the TNL and partly in the APD, leading to all sensitivity runs being independent of *r* since, in the calculation of Q, both noise bandwidth-adoption methods will then cancel, with an increase in the received signal power (solid lines in Figure 10). When adding a fixed *I_dm_* = 2.5 nA, however (dashed lines in Figure 10), its changing ratio vs. the received power will again lead to *r* dependency, and we see a strong performance degradation specifically with SNL due to the false detection of electrons from *I_dm_*. For *TNL,* however, the plot against *I_dm_* is too close to the original curve to be distinguishable (red line(s)).

In practical systems we might see both RFE characteristics; when both *B* and the *R_TI_* can be optimized to one fixed *r*, the sensitivity improves and so does the constant for Ppb (Figure 10). When, however, in a variable-rate system, only the bandwidth of the Rx filter can be adopted to *r*, the overall sensitivity will reduce with a decreasing *B* when thermal noise is present, as shown in Figure 9.

## 5. Optimum *M* with Background Light

### 5.1. Power of Background Light

In an FSO receiver, background light from the sun or sunlight reflected by celestial bodies or clouds, air, and ground structures, are significant sources of noise and electronic offsets for detectors. BGL-induced current flowing through the APD will add extra shot-noise, just as the detector’s dark current does.

Values of *P_BGL_* depend on the receiver’s aperture size, the field-of-view (FoV) angle of the detector, the optical bandwidth of background blocking filters, and the spectral behavior of the background source itself, which also implies the wavelength (e.g., blue sky and haze at certain elevations, sunlit clouds, and spectral luminance of the background structure behind a data source) [28,29,30,31,32]. These environmental parameters can be summarized by the spectral irradiance, *L_e,Ω,λ_*, in [W/m^2^/nm/sr], see Table 1. We only state here some of the typical figures regarding the sun far above the horizon to allow a rough estimation of *P_BGL_*, where a detailed analysis would be required for the precise values in individual scenarios (more precisely, the angle of the sun-earth target and the sun elevation would have to be considered):

Small celestial bodies (like planets or stars) exhibit a very limited apparent size which normally falls below the detector’s FoV. Looking into the sun must be absolutely avoided in any case.
(22)$$P_{bgl}=L_{e,\Omega ,\lambda }\cdot A\cdot \Omega \cdot \Delta \lambda$$
where the receiver-telescope aperture area, *A*, is in m^2^, the optical filter bandwidth, Δ*λ*, is in nm, and the detector’s solid angle FoV, *Ω*, is derived from the more common full-flat angle *ω*: *Ω =* 4π sin^2^ω (Table 2).

With typical OGS geometries, and for a 1550 nm signal wavelength, we find the following exemplary values for the background light:

**Table 2 sensors-22-06773-t002:** Typical values of *P_BGL_* for a 1550 nm wavelength.

Scene	$L_{e,\Omega ,\lambda }$ in W/(m^2^ nm sr)	*A* in m^2^	*ω* in µrad	Δλ in nm	*P_BGL_* in nW
towards horizon	25 × 10^−3^	0.1	200	40	50
towards horizon	25 × 10^−3^	0.7	100	40	88
to zenith	1.2 × 10^−3^	0.1	200	40	2.4
to zenith	1.2 × 10^−3^	0.7	100	40	4.2

Where 0.1 m^2^ refers approximately to a Cassegrain telescope with a 40 cm aperture diameter and 0.7 m^2^ to a 1 m telescope. A 40 nm-wide filter covers the whole C-Band and thus supports the span of several DWDM channels.

We find that 88 nW is a large value for *P_BGL_* from the horizon sky during daytime (and a similar value when looking onto the sun-illuminated moon disk), while 2.4 nW serves as a minimum value. At nighttime, of course, BGL should be negligible (except when celestial bodies are in the FoV). For further examples, we chose 50 nW as a typical strong BGL value.

### 5.2. Optimum Multiplication Factor with BGL

From the above analysis (19), we compare the BER with *M*, without and with BGL.

The optimum multiplication factor for the examples in Figure 11 varies from ~10 to ~40, and a wrong value can reduce the BER by orders of magnitude, e.g., with a 50 nW BGL and a 200 Ppb *M_opt_* = 12, whereas, without BGL the *M_opt_* = 28, and using the optimum *M* from before would increase the BER from 10^−8^ (at *M =* 28) to 10^−6^. Controlling *M* dependent on the varying *<P_Rx_>* and BGL, therefore, is a must in optimized FSO-APD-receivers.

As with (7), from (19) we can derive an optimum *M* for the maximum *Q* for APD data receivers, with the exponential approximation of *F_A_* and with NRZ modulation. Now, again, regarding *I_dm_* and BGL:(23)$$\begin{array}{l} \frac{dQ}{dM}=0\;\to d\left(\frac{M}{\sqrt{M^{2+x}A+B}+\sqrt{M^{2+x}\left(A+C\right)+B}}\right)/dM=0\\ with\;A=2q\left(RP_{bgl}+I_{dm}\right)\;;\;B=i_{t}{}^{2}\;;\;C=2qR\cdot 2\langle P_{Rx}\rangle \;;\;x=k_{A}{}^{0.355} \end{array}$$

In contrast to the term with CW illumination, the derivation here results in:(24)$$M_{opt}={\left(\frac{-b+\sqrt{b^{2}-4ac}}{2a}\right)}^{\frac{1}{2+x}}$$
with



$\begin{array}{l} a=x^{2}AC(A+C)\\ b=x^{2}BC(2A+C)\\ c=-4(x+1)B^{2}C \end{array}$



As shown in Figure 12 for different *k_A_*, *M_opt_* with data modulation differs significantly from CW illumination (compare Figure 3).

The relationship in Chapter 4.5.2 of [45], and from [46], also describe the relation for an optimum *M*; however, this requires numerical evaluation.

For the following comparisons, we assume the values from Table 2 with a 1 Gbps data rate. The optimization of *M* is independent of the data rate (which only requires the fraction of on-time vs. off-time to be 50:50) but depends on the received signal power *<P_Rx_>*. When regarding bit length, however, we can relate this received power to energy per bit.

With the optimization of *M* from (23) and (24), we understand the importance of adjusting *M* to the received power, especially for the low-power/high-BER regime. Figure 13 shows the effect of the optimized vs. fixed *M* on the BER, with and without background light and dark current.

A fixed and high *M =* 25 offers comparable performance to the optimized *M* when no BGL and no *I_dm_* are present, while a low *M =* 8 shows inferior sensitivity. But the same fixed high *M* will provide less sensitivity when these additional shot noise sources are present, and the low *M =* 8 then nearly coincides with the optimized *M*. In any case, the curves are much closer to each other when additional shot-noise is present.

Methods to optimize *M* automatically for varying the input power have been suggested for fiber communications [55]. Such methods, however, are of marginal applicability in the case of FSO, with its large and variable fraction of background light and with an even faster varying received power (due to scintillation and point-fading). Rather, a BGL sensor would need to be added and evaluated to ensure a minimum *p_BE_* for any situation. Thus, when BGL and *I_dm_* can be measured and are not negligible, it is important to ensure the advantageous fixed multiplication factor for an individual environmental situation. Figure 14 signifies its importance by plotting the BER and the *M_opt_* over a range of BGL and three values of *<P_Rx_>* (no *I_dm_* is regarded here for better comparability).

We find that the multiplication factor, *M_opt_*, requires careful adoption to the instantaneous background light to ensure the optimum performance of the APD-RFE, while the influence of absolute received power on *M_opt_* is less significant in this example since its shot-noise component is small compared to BGL. The more background light, the less important the individual optimization of *M*, tending towards a value around 8 with the parametrization used here.

These foregone findings are verified by measurements with free-space APD-RFE-implementation, with and without the influence of a 1550 nm BGL source (Figure 15).

The practical RFE measured (Figure 15) shows some real-world deteriorations; the bandwidth of TIA along with the reception lowpass (LP) and limiter does not match, and the high capacitance of the large APD (200 µm in diameter) again limits its usable data rate and, thus, its sensitivity. These lead to a 320 Ppb when BER = 10^−4^ instead of the ideal ~120 Ppb. The sensitivity run and the effect of the background light coincide well with the predictions from the formalism presented in this section, with a deterioration in sensitivity of nearly −3 dB when BER = 10^−4^, with and without BGL.

## 6. Summary and Conclusions

In this document, we summarize the basic relations of free-space APD receivers with an emphasis on optical LEO data downlinks. Such data reception is prone to fast and strong signal- and background-light variations due to atmospheric and mechanical effects. Furthermore, changes in atmospheric attenuation and the strength of the background light with elevation add to this dynamic parametrization. We derive a model for the optimum multiplication factor and evaluate its relation to signal power, background light, and other parameters.

We find that the dependence of the optimum multiplication factor from the background light and data signal strength suggests that both parameters need to be measured. Accordingly, the control of the APD biasing-voltage, *U_R_*, is recommended to ensure optimum receiver performance in all FSO situations, in addition to its dependency on temperature. With a high amount of background, the optimization of the multiplication becomes less important, and rather a fixed value can be used.

The RFE performance also suffers from other effects, like the non-matching bandwidths of TIA, reception filters and limiters, and noise effects from the limited stability of the biasing voltage.

One of the parameters that can be practically controlled is the limitation of the background light through denser chromatic filtering; however, the data signal’s channel width (spectral broadness of data signal) sets a limit to this reduction. Furthermore, the receiver telescope’s FoV can be reduced by opto-mechanic measures, with this then requiring more precise pointing and tracking regarding the signal source during communication. Other parameters lie in the sophistication of the APD itself, namely the tapering of dark current and excess noise factors.

## Figures and Tables

**Figure 1 sensors-22-06773-f001:**
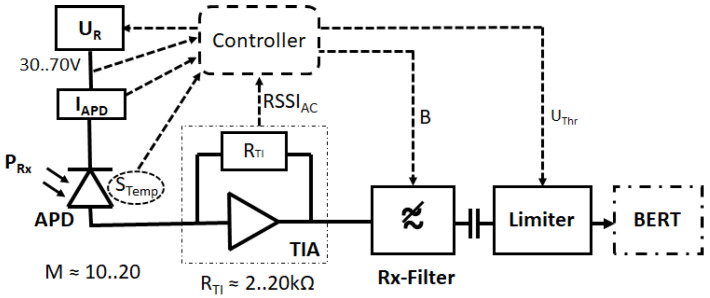
Practical InGaAs-APD receiver diagram for high-speed free-space optical (FSO) communications.

**Figure 2 sensors-22-06773-f002:**
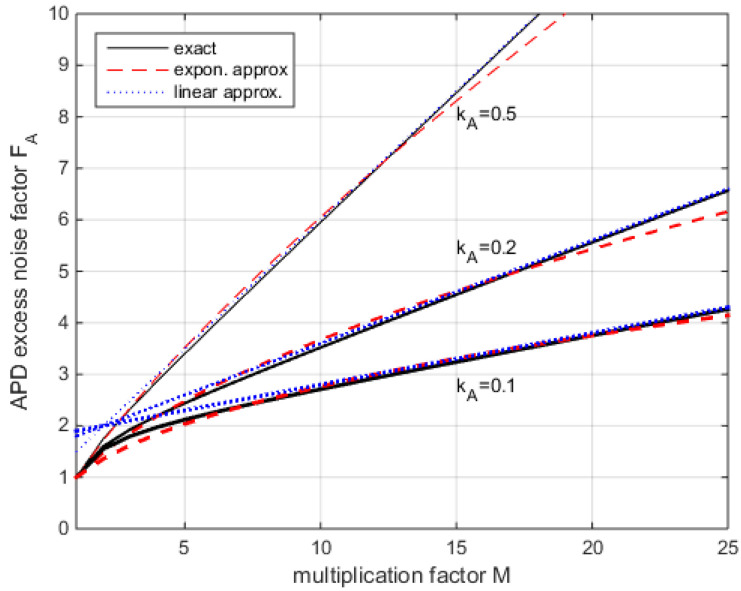
Comparison of approximations (5) and (6) with exact solution (4).

**Figure 3 sensors-22-06773-f003:**
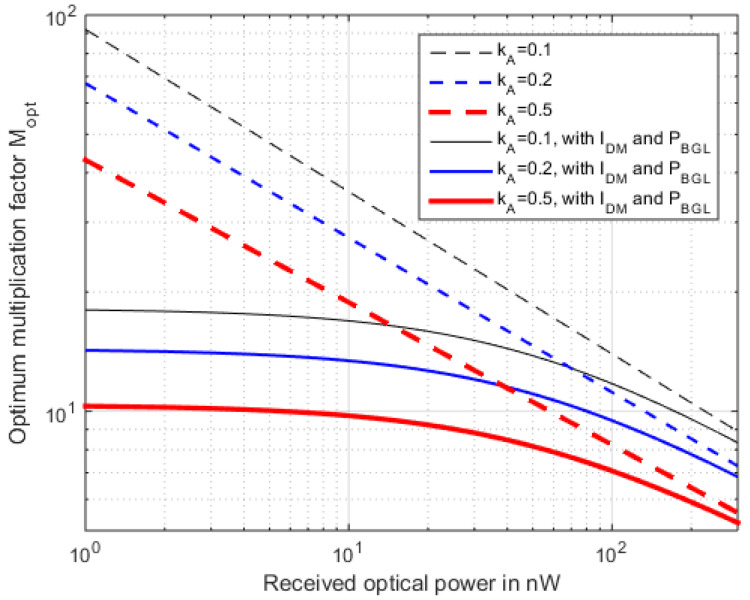
Optimum multiplication factor *M_opt_* with CW illumination, for different *k_A_*, without and with *I_dm_* = 2.5 nA and *P_BGL_* = 50 nW.

**Figure 4 sensors-22-06773-f004:**
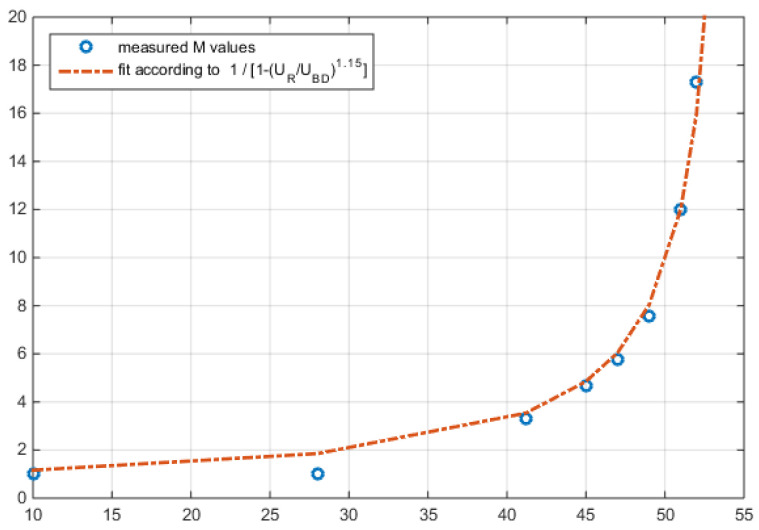
Measured and calculated *M*, according to (8), over the reverse voltage *U_R_* for *U_BD_* = 55 V and *n* = 1.15.

**Figure 5 sensors-22-06773-f005:**
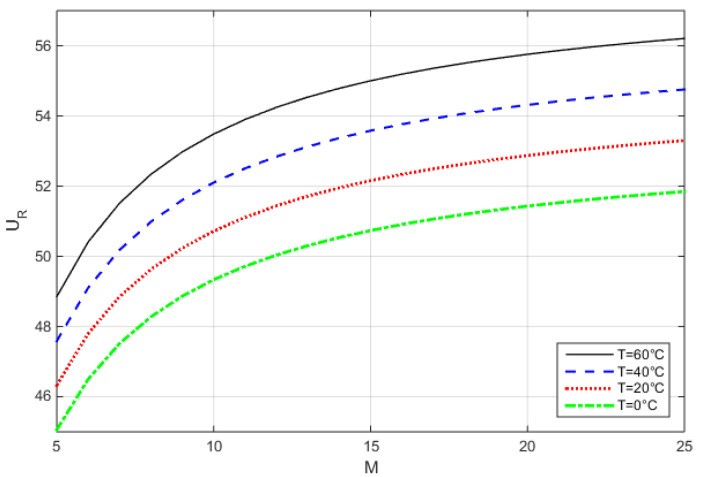
*U_R_(M,T)* for *U_BDref_* = 55 V at 20 °C, and *ρ_T_* = 0.075 V/°C.

**Figure 6 sensors-22-06773-f006:**
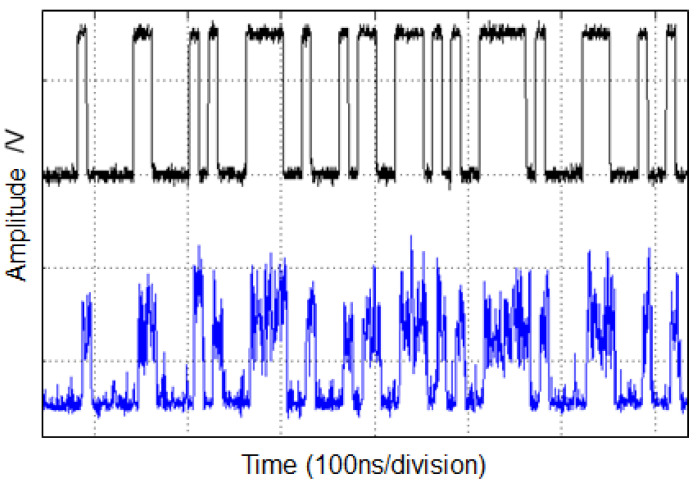
Typical binary OOK data bit stream at 100 Mbps, as monitored by an oscilloscope: upper plot transmitted data, lower plot is the received signal from an APD-RFE.

**Figure 7 sensors-22-06773-f007:**
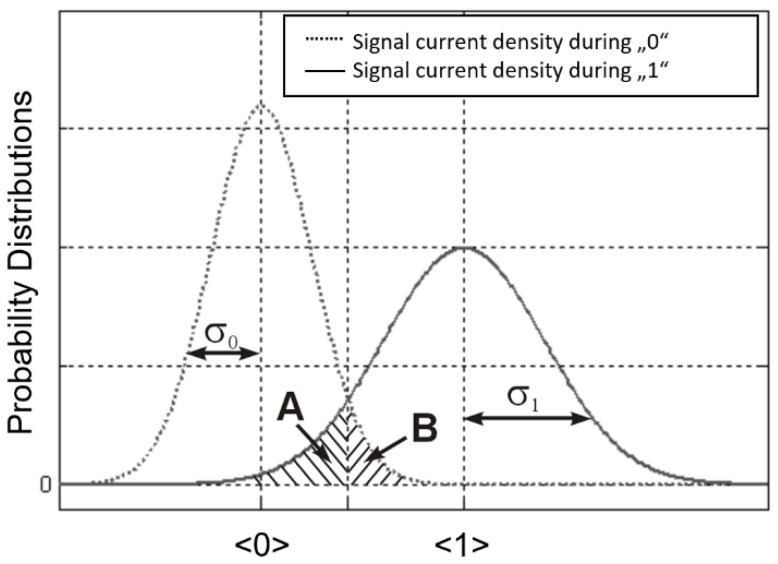
Generic situation of the Gaussian-shaped noise distributions at binary 1 and 0.

**Figure 8 sensors-22-06773-f008:**
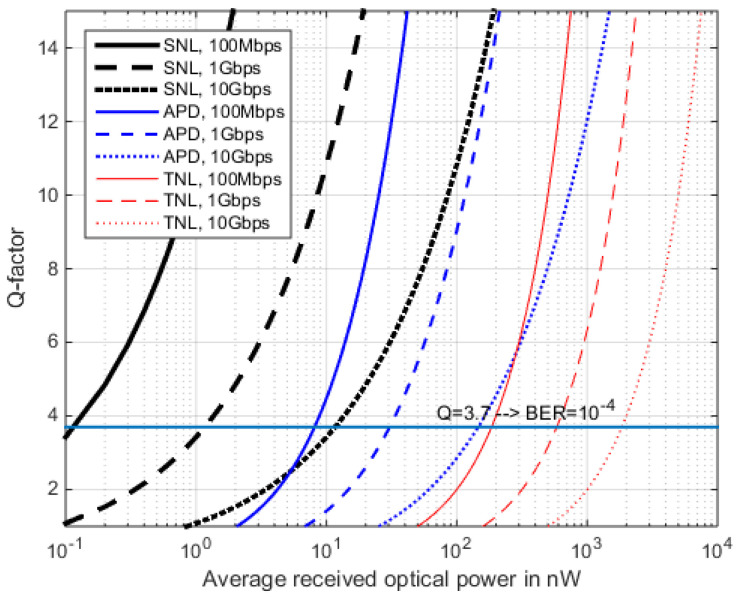
RFE sensitivity for the three RFE cases where *M* = 25. The RFE here has a constant TIA noise density but an adoptive Rx lowpass.

**Figure 9 sensors-22-06773-f009:**
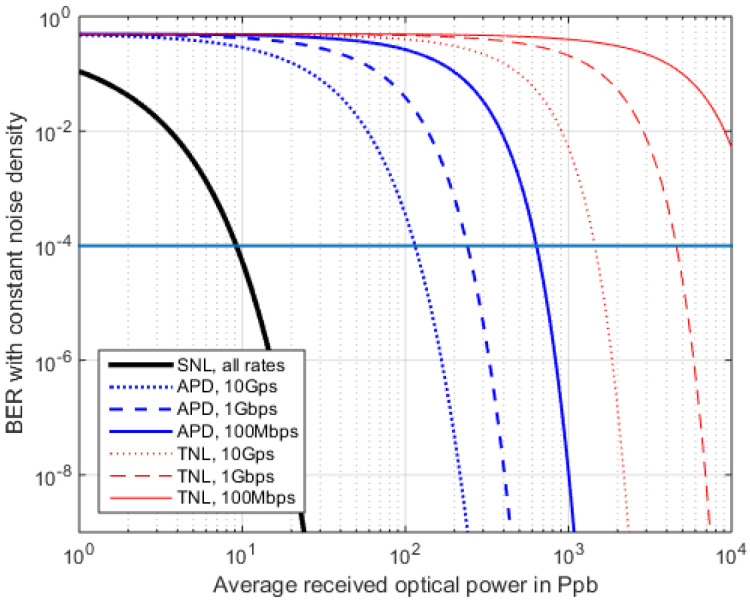
BER from *<N>*, with a constant TIA noise density and adaption of receiver filter only. Same parametrization as Figure 8.

**Figure 10 sensors-22-06773-f010:**
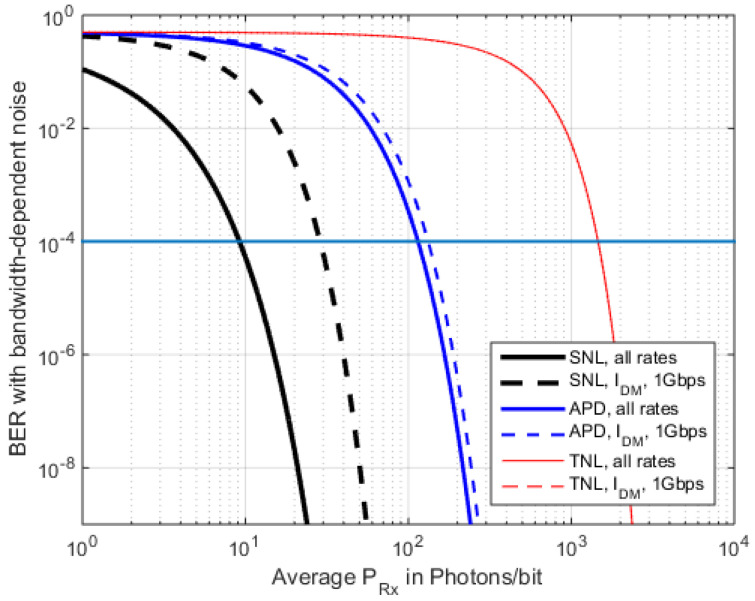
BER from Ppb for a sqrt(*B*)-dependent TIA noise density, according to the data rate, plus the adaptation of the reception filter bandwidth. Sensitivity curves with *I_dm_* are shown for comparison (for 1 Gbps, no BGL).

**Figure 11 sensors-22-06773-f011:**
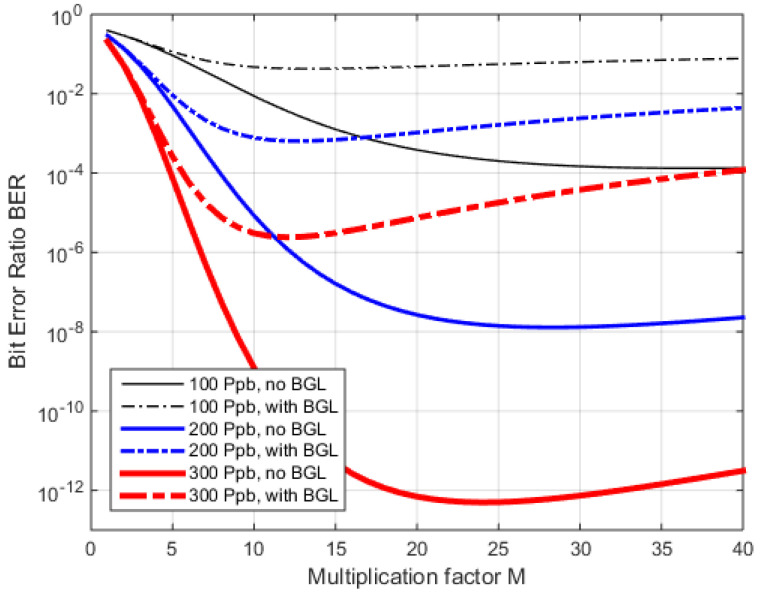
Dependency of BER against *M*, without and with BGL (*P_BGL_* = 50 nW), and three received signal levels for 1 Gbps.

**Figure 12 sensors-22-06773-f012:**
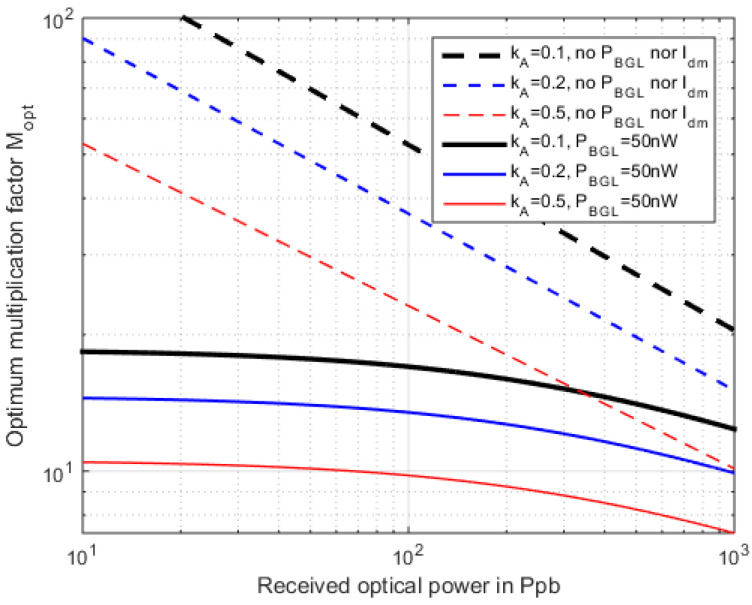
Optimum multiplication factor *M_opt_* with an NRZ-50:50 modulated signal for different ionization ratios, according to (24).

**Figure 13 sensors-22-06773-f013:**
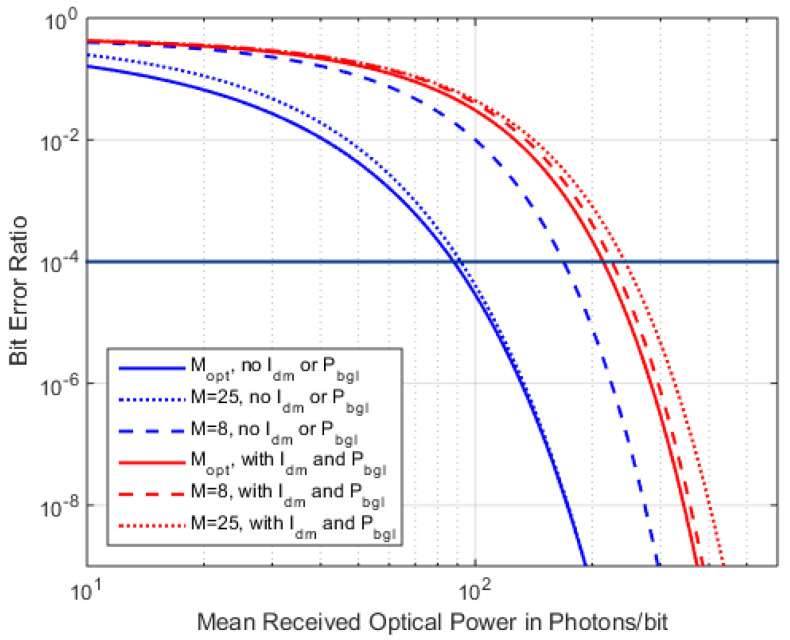
BER over *<N>*, with a fixed *M* = 8 and *M* = 25 compared with *M_opt_* adopted to *P_Rx_*, without and with *P_BGL_* = 50 nW and *I_dm_* = 2.5 nA.

**Figure 14 sensors-22-06773-f014:**
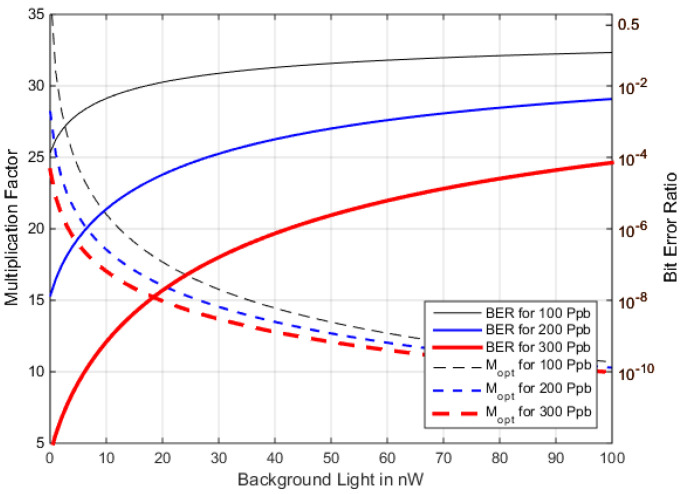
BER and *M_opt_* over *P_BGL_* at a *P_Rx_* of 100/200/300 Ppb, for BGL from 0 to 100 nW (no *I_dm_* to visualize the effect of BGL only).

**Figure 15 sensors-22-06773-f015:**
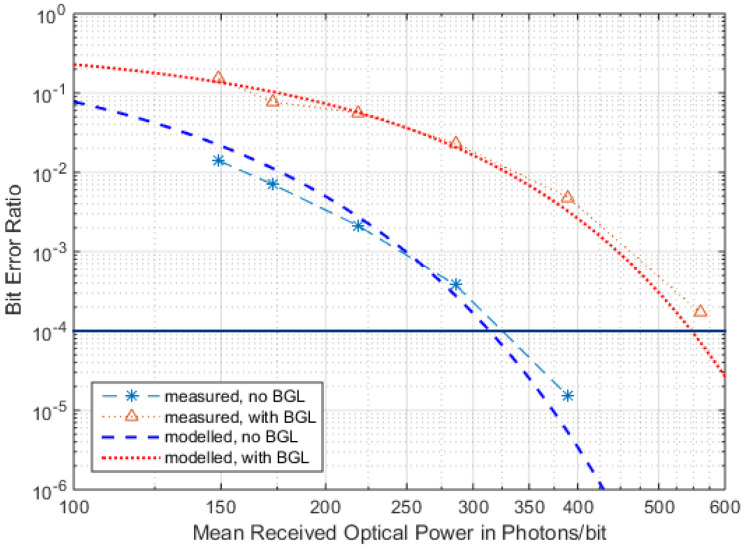
Measured BER performance with and without BGL for *M_opt_* (*i_n_* = 2.5 pA/Hz^0.5^, *I_dm_* = 9 nW, *P_BGL_* = 50 nW, *B* = 500 MHz, *r* = 300 Mbps).

**Table 1 sensors-22-06773-t001:** BGL spectral irradiance at different wavelengths.

W/(m^2^ nm sr)	*λ* = 850 nm	*λ* = 1064 nm	*λ* = 1550 nm
on Sun-disk	20 × 10^3^	10 × 10^3^	2 × 10^3^
blue sky zenith	2.0 × 10^−3^	2.3 × 10^−3^	1.2 × 10^−3^
blue sky 30° el.	3.5 × 10^−3^	4.0 × 10^−3^	2.0 × 10^−3^
blue sky horizon	30 × 10^−3^	30 × 10^−3^	25 × 10^−3^
sunlit cloud	200 × 10^−3^	80 × 10^−3^	20 × 10^−3^
overcast cloud	20 × 10^−3^	8 × 10^−3^	2 × 10^−3^
on Moon-disk	400 × 10^−3^	220 × 10^−3^	20 × 10^−3^
full Mars-disk	11 × 10^−12^	8 × 10^−12^	3 × 10^−12^

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
