# Peer review of "Free-Space Optical Data Receivers with Avalanche Detectors for Satellite Downlinks Regarding Background Light"

_sensors, 2022, doi:10.3390/s22186773_

Round 1

Reviewer 1 Report

This manuscript explains why the precise controls such as high-voltage, temperature, back-ground light, and subsequent transimpedance, has to be satisfied for using the avalanche detectors in the free-space-optical data receivers. The receiver performance with background light is also experimentally verified by the authors. This manuscript clearly tests the functions of avalanche detector and optimizes some multiplication factors. This manuscript can be published in the journal Sensors with some typos corrected.

Comments:                                                                                                          

1.      Line 138, “and thermal noise density it [A/sqrt(Hz)]”  à  “and thermal noise density it [A/ ]”.

2.      Line 221, “shows a typical PRBS (pseudo-random bit sequence) with”  à  “shows a typical pseudo-random bit sequence (PRBS) with”.

3.      Line 225, please add the “unit” to the vertical axis “Amplitude”.

4.      Line 281, “(compare Fig. 11)”  à  “(compare figure 11)” .

Line 527, The page numbers were missing in many references, e.g. ref. 5, ref. 8, …. 

Author Response

For information to the Editor, and individual Reviewer: I did use the original two-column .doc document for the revisions, since the version from MDPI, as automatically converted during primary submission, did not show the line-numbers as used by some reviewers, and also (very few) formula-disambiguates seem to be introduced through the conversion-process. This might have happened by my use of different formula editors and I hope we can align this during further processing.

- I deleted a spare picture of a formula in front of formula (3).
- Any changes during the revision are marked in red text color.
- Some slight rephrasing was done in the text and marked in red.

Sincerely, Dirk Giggenbach

REVIEWER 1: 

Comments:                                                                                                          

  1. Line 138, “and thermal noise density it [A/sqrt(Hz)]”  à “and thermal noise density it [A/ ]”.
    Answer: I am not sure if I understood correctly – we could replace “sqrt” by the root-sign…
  2. Line 221, “shows a typical PRBS (pseudo-random bit sequence) with”  à “shows a typical pseudo-random bit sequence (PRBS) with”.
    A: done
  3. Line 225, please add the “unit” to the vertical axis “Amplitude”.
    A: done
  4. Line 281, “(compare Fig. 11)”  à “(compare figure 11)” .
    A: done

Line 527, The page numbers were missing in many references, e.g. ref. 5, ref. 8, …. 
A: page numbers were often omitted in the refernces since in online paper databases these seem unnecessary, so I try to reduce their use. Maybe MDPI can advise on how to handle this.

Reviewer 2 Report

Review of tutorial paper on Free-space optical data receivers with Avalanche detectors for satellite downlinks regarding background light from D. Giggenback

I found this to be an interesting tutorial paper from my perspective, which is that of a university professor teaching optical fiber communications. My teachings deal with photodetection, also addressing the case (and advantages) of using avalanche photodetectors. The particular issues faced for satellite downlinks and the tradeoffs were interesting and this is a worthy paper to publish. I do have a few questions/recommendations, however, that can improve the paper for its target audience.

1.       I would welcome some more general information on ground to satellite links, and here a little graphic would be very welcome. Satellite downlinks can be of low, mid, high and geostationary earth orbits. What are the different implications of each? For low earth orbit, I assume the tracking must be very agile. Can the author provide some information? Also, the author mentions that looking into the sun must be absolutely avoided. Is this feasible when tracking satellites, or is the link broken during the time satellite crosses between the receiver and sun? Why is the background light greater for horizon viewing over zenith? Is it due to less Rayleigh scattering (from a smaller effective volume of atmosphere in view)? Author mentions that modern systems utilize 1550 nm over 850nm. I presume this is due to lesser background light, as shown in Table 1, but this is never stated explicitely. Also, Si-based avalanche PDs achieve higher gain factors M over InGaAs. Is there a tradeoff here and is background light the sole reason for the shift to 1550 nm?

2.       The paper discusses satellite downlinks, but it would be of interest to find out whether the uplinks have many of the same issues. Do the satellite’s also use avalanche PD? Is scattered light from the earth’s ground a severe source of background light?

3.       The performance is evaluated at different baud rates, suggesting adaptive baud rates are employed (not common in the more predictable fiber-optic channel). How important is this feature? Also, the impact of the electric bandwidth is assessed, especially when it is not adaptive together with the bitrate (it is mentioned that it is difficult to make it adaptive). Is it not beneficial to complicate the ground-based RFE to squeeze out all performance? Is it not possible to adaptively control the gain, M, using error counts to keep the BER at minimum?

4.       The paper is very “link engineering” oriented. I would welcome a little extra device physics information (on the avalanche PD), especially in Sections 2 and 3.

5.       Some mistakes that need to be addressed:

a.       At least in the PDF I see on my computer, there’s a typo or character problem in Eq. 2 (I see a question mark between R and P_bgl in the denominator).

b.       Line 194, the word ‘were’ should probably be ‘where’

c.       Line 211, the word ‘analog’ should probably be ‘along’

d.       Line 375 suggests comparing Figure 12 with figure 5. Likely the figure to compare with is Fig 3.

e.       Line 429, the word ‘prawn’ (as is shrimp) should likely be “prone”

Author Response

For information to the Editor, and individual Reviewer: I did use the original two-column .doc document for the revisions, since the version from MDPI, as automatically converted during primary submission, did not show the line-numbers as used by some reviewers, and also (very few) formula-disambiguates seem to be introduced through the conversion-process. This might have happened by my use of different formula editors and I hope we can align this during further processing.
- I deleted a spare picture of a formula in front of formula (3).
- Any changes during the revision are marked in red text color.
- Some slight rephrasing was done in the text and marked in red.
Sincerely, Dirk Giggenbach

  1. I would welcome some more general information on ground to satellite links, and here a little graphic would be very welcome. Satellite downlinks can be of low, mid, high and geostationary earth orbits. What are the different implications of each? For low earth orbit, I assume the tracking must be very agile. Can the author provide some information? Also, the author mentions that looking into the sun must be absolutely avoided. Is this feasible when tracking satellites, or is the link broken during the time satellite crosses between the receiver and sun? Why is the background light greater for horizon viewing over zenith? Is it due to less Rayleigh scattering (from a smaller effective volume of atmosphere in view)? Author mentions that modern systems utilize 1550 nm over 850nm. I presume this is due to lesser background light, as shown in Table 1, but this is never stated explicitely. Also, Si-based avalanche PDs achieve higher gain factors M over InGaAs. Is there a tradeoff here and is background light the sole reason for the shift to 1550 nm?

    A: This paper being specific on the optical LEO-downlink – as described in the Introduction – I would refer to the references for describing this application scenario.

    I added some explanations for the wavelength-selection changing from 8xx nm to 15xx nm. Background light plays a negligible part, but component availability and in LEO-downlinks the reduced atmospheric attenuation.

    A sentence on the fast LEO movement was added.

    The Sun in the background of an optical space-link would overexpose the receiving telescope and its Direct-Detection tracking sensors beyond spec, probably destroying it. Coherent BPSK homodyne systems claim they could track through the sun, but would not demonstrate it to not endanger their systems… Anyway the sun behind the communication partner happens very seldom (its area is ~1/5E5 of the sky’s area), and a link can just avoid this geometry.

    The atmospheric scattering increases towards the horizon since more low-height atmospheric volume is traversed, which also holds more aerosols (sun-reflecting as well as attenuating). Rayleigh-scattering is not so relevant at 1550 and also 850. Please refer to this publication for details:
    D. Giggenbach, A. Shrestha, “Atmospheric absorption and scattering impact on optical satellite-ground links”, Int J Satell Commun Network. 2021;1-20, 6 Oct.2021, doi:10.1002/sat.1426

  2. The paper discusses satellite downlinks, but it would be of interest to find out whether the uplinks have many of the same issues. Do the satellite’s also use avalanche PD? Is scattered light from the earth’s ground a severe source of background light?

    A: Uplink is not discussed here since an APD-receiver on a satellite would have to deal with other challenges, like radiation and maybe extended temperature-range, etc. . The application decides if APDs are needed (data rate and pointing-quality), or if simplex PIN-receivers are sufficient, or if even coherent high-speed DWDM systems are required (mostly to GEO). Also, the asymmetric IRT-channel will lead to more fading in the uplink than in the downlink, since no aperture averaging can be applied for uplinks, just transmitter-diversity. Again, since this paper only evaluates APD-RFEs for optical ground stations, only the downlink is considered. Scattered light albedo from Earth ground is not considered.
    I recommend the following for ground-space links:
    M. Knopp, A. Spörl, M Gnat, G. Rossmanith, F. Huber, C. Fuchs, D. Giggenbach “Towards the utilization of optical ground-to-space links for low earth orbiting spacecraft”, Acta Astronautica (166), pages 147-155. Elsevier
    M. Knopp, D. Giggenbach, R. Mata Calvo, C. Fuchs, K. Saucke, F. Heine, F. Sellmaier, F. Huber, „Connectivity services based on optical ground-to-space links”, Acta Astronautica, 148. Elsevier. ISBN 0094-5765, ISSN 0094-5765, April 2018

  1. The performance is evaluated at different baud rates, suggesting adaptive baud rates are employed (not common in the more predictable fiber-optic channel). How important is this feature? Also, the impact of the electric bandwidth is assessed, especially when it is not adaptive together with the bitrate (it is mentioned that it is difficult to make it adaptive). Is it not beneficial to complicate the ground-based RFE to squeeze out all performance? Is it not possible to adaptively control the gain, M, using error counts to keep the BER at minimum?
    A: Yes, adaptive data rate is important to use in LEO-downlinks as the received power changes by more than factor 20 from 5° elevation to zenith and back. Variable data rate also became part of the standardization in CCSDS. Different methods to allow improved sensitivity at different data rates are followed by implementers currently, where the level of “shot-noise-limitedness” of the RFE decides about the necessity to adopt the bandwidth. We will see in future implementations of Sat-FSO-RFEs if squeezing out the best bandwidth is necessary, or if other receiver-types require different adaptation.
    B. Edwards, K-J. Schulz, J. Hamkins, B. Robinson, R. Daddato, R. Alliss, D. Giggenbach, L. Braatz, “An Update on the CCSDS Optical Communications Working Group”, IEEE-Xplore, International Conference on Space Optical Systems 2019 (ICSOS), 14.-16. Oct 2019, Portland, USA
    P.D. Arapoglou, N. Mazzali, “System-level Benefit of Variable Data Rate in Optical LEO Direct-to Earth Links”, International Conference on Space Optics ICSO 2020

  2. The paper is very “link engineering” oriented. I would welcome a little extra device physics information (on the avalanche PD), especially in Sections 2 and 3.
    A: Some sentences explaining APDs’ principle functioning were added at section II.

  3. Some mistakes that need to be addressed:
    a) At least in the PDF I see on my computer, there’s a typo or character problem in Eq. 2 (I see a question mark between R and P_bgl in the denominator).
    A: this must be a conversion problem: I submitted the paper in a 2-column format, maybe using different versions of a formula editor. The character you are referring to should be a multiplication-dot I only inserted for clarity, I now removed it.
    b) Line 194, the word ‘were’ should probably be ‘where’
    A: correct, I changed to ‘where’
    c) Line 211, the word ‘analog’ should probably be ‘along’
    A: yes that sounds better, I changed to ‘along’
    d) Line 375 suggests comparing Figure 12 with figure 5. Likely the figure to compare with is Fig 3.
    A: yes, corrected to figure 3
    e) Line 429, the word ‘prawn’ (as is shrimp) should likely be “prone”
    A: yes, corrected

Reviewer 3 Report

The manuscript “Free-Space-Optical Data Receivers with Avalanche Detectors 2 for Satellite-Downlinks regarding Background Light” by Dirk Giggenbach is devoted to both theoretical and practical aspects of the usage of avalanche photodiodes for free-space optical communication.

The paper is written in a way that is accessible to non-professionals and in the same time contains useful information for actually building circuits and using the APDs for light detection. This style is very suitable for the tutorial.

In particular, the author gives practically-related information (can be found in other sources but not easily): how exactly the internal (avalanche) multiplication factor is related with bias voltage and temperature; what gain is optimum for CW illumination; what is the influence of dark current and other parameters on the quality of data transmission, etc. All formulae are discussed in terms of simple physical intuition. Finally, the model is derived for the optimum multiplication factor and its relation to signal power, background light, and other parameters is evaluated.

One disadvantage of the paper is a lot of abbreviations, sometimes unexplained and probably unnecessary. Some of the actually impede fast reading, for instance: “intensity-modulation and direct detection (IM/DD)”, “adaptive optics (AO)” are defined and not used. Other are used without definition, like “QKD” and “LEO”. Probably these terms are well-known in space communication field, but many potential readers of the paper are far from this area but use avalanche photodiodes as well. Thus, I believe that keeping the number of abbreviations as low as possible would greatly improve the text.

Minor points:

The very first letter in the introduction is omitted (VALANCHE instead of AVALANCHE)

Lines 124, 125: please provide the names of variables

Eq. (5) seems very unusual. Is it original result? Otherwise could the author provide a reference?

Line 184: -.5 in the exponent should be -0.5

Line 77: redundant “]”

Line 278: link to the table II probably should be to table 3?

Some words seem to be not suitable:

Line 132: “according” – did the author mean “corresponding”?

Line 147: “stated” – probably “related”

Line 187: “proceeding” – probably “assumption”

Author Response

For information to the Editor, and individual Reviewer: I did use the original two-column .doc document for the revisions, since the version from MDPI, as automatically converted during primary submission, did not show the line-numbers as used by some reviewers, and also (very few) formula-disambiguates seem to be introduced through the conversion-process. This might have happened by my use of different formula editors and I hope we can align this during further processing.
- I deleted a spare picture of a formula in front of formula (3).
- Any changes during the revision are marked in red text color.
- Some slight rephrasing was done in the text and marked in red.
Sincerely, Dirk Giggenbach

Reviewer 3:

... One disadvantage of the paper is a lot of abbreviations, sometimes unexplained and probably unnecessary. Some of the actually impede fast reading, for instance: “intensity-modulation and direct detection (IM/DD)”, “adaptive optics (AO)” are defined and not used. Other are used without definition, like “QKD” and “LEO”. Probably these terms are well-known in space communication field, but many potential readers of the paper are far from this area but use avalanche photodiodes as well. Thus, I believe that keeping the number of abbreviations as low as possible would greatly improve the text.  

A:
Dear Reviewer, thank you for your valuable advices. I will have the text checked by MDPI Language Editing Service to improve further. This can however only be done after my submission of this revision, due to time-constraints.

A: AO is not further used and I deleted its abbreviation. IM/DD has however a wide-spread use in optical satellite communications – in contrast to coherent modulation formats – and it is used twice in the text, so the abbreviation should be kept.       
QKD has been deleted, and LEO is now explained. More three-letter acronyms have been removed.        

Minor points:

The very first letter in the introduction is omitted (VALANCHE instead of AVALANCHE)
A: this must be a conversion-error during submission, it appears as an enlarged letter in my original submission. Maybe it can be cared for by the MDPI-editor ?

Lines 124, 125: please provide the names of variables       
A: I am not sure if I can identify the right spot in the text since my word generates different line numbers, but I guess you are referring to: “…with a load resistor RL and the amplifier’s noise figure Fn as Fn·4kBT/RL. False background light power Pbgl can…”. I changed the sentence, and added to the “Acronyms and Symbols”.

Eq. (5) seems very unusual. Is it original result? Otherwise could the author provide a reference?
A: the approximation stems from [42] {Application Note, “Avalanche photodiodes – A User Guide”, PerkinElmer Optoelectronics, 2010} where I used as exponent the “0.355” as it most fit to the experimental data. The reference [42] is stated three lines below.

Line 184: -.5 in the exponent should be -0.5
A: Okay, corrected

Line 77: redundant “]”          
A: Okay, corrected

Line 278: link to the table II probably should be to table 3?           
A: Okay,corrected

Some words seem to be not suitable:

Line 132: “according” – did the author mean “corresponding”?    
A: sorry, due to missing line numbers I could not identify which of my many “according”s is meant here, I hope for the proof-reading by MDPI…

Line 147: “stated” – probably “related”      
A: same as above

Line 187: “proceeding” – probably “assumption”  
A: Okay, identified and changed
